# Does environmental management system certification keep enterprises out of trouble? Evidence from stock price crash risk

**Hongyu Liu**[1], **Qin Binbin**[2], **Pengliang Qiao**[1] *

1 School of Management, Guangzhou College of Technology and Business, Foshan City, China, 2 School of Business Administration, South China University of Technology (SCUT), Guangzhou, China

* qpliang@gzgs.edu.cn

**Data Availability Statement:** All relevant data are within the manuscript and its Supporting Information files.

## Abstract

This paper examines the impact of environmental management system (EMS) certification, a significant voluntary participatory environmental regulation, on the risk of stock price collapse. The study is based on sample data of heavily polluting listed companies from 2008–2020. The study demonstrates that certification of environmental management systems has a significant impact on preventing share price collapse. This finding remains consistent even after controlling for endogeneity and conducting robustness tests. The analysis also reveals that the inhibitory effect of EMS certification is more pronounced for state-owned enterprises and firms with a higher degree of marketisation. Exploring the mechanism of its influence, it is found that environmental management system certification mainly suppresses the risk of stock price collapse by improving the environmental performance of enterprises and the transparency of corporate information, suggesting that environmental management system certification can be used as both an "environmental governance tool" for suppressing stock price collapse and an "information transfer tool" for improving the transparency of corporate information, thus suppressing the risk of stock price collapse. Meanwhile, the media's attention has been found to moderate the effect of environmental management system certification on stock price crash risk. These findings validate the inhibitory effect of environmental management system certification on stock price crash risk, expand our understanding of the economic consequences of environmental management system certification and the factors that influence stock price crash risk. They also provide a theoretical basis and practical support for environmental regulators.

## 1. Introduction

The International Organization for Standardization (ISO) introduced the Environmental Management System (EMS) specification, ISO 14001, in 1996, marking a significant milestone in corporate environmental management and sustainable development [1]. Despite its importance, there remains a gap in systematic research exploring how this specification influences stock price crash risk [2]. Understanding this mechanism is crucial for enterprises to

**Funding:** This research was funded by National Natural Science Foundation of China (NSFC No. 71962005), Innovation Team Project for Ordinary Universities in Guangdong Province (Humanities and Social Sciences) (2023WCXTD026) and Major Scientific Research Projects in Colleges and Universities in Guangdong (Grant No. 2021ZDJS122)(Grant No. 2022ZDJS142). Guangdong Province Philosophy and Social Science Planning Youth Project (GD23YGL28) Pengliang Qiao and Hongyu Liu, as funders, were involved in the design of the study, data collection, analysis, interpretation of data, and writing of the manuscript.

**Competing interests:** The authors have declared that no competing interests exist.

effectively implement ISO 14001, thereby improving their environmental performance and gaining public trust and recognition.

China adopted ISO 14001 as a national standard in 1996, establishing its own EMS certification system to regulate environmental practices in enterprises and organizations. As a voluntary environmental regulation, complying with ISO 14001 has become a key indicator of a company's 'green image,' social responsibility, and public trust. Adherence to this standard involves setting up a comprehensive EMS, enhancing environmental performance, reducing risk, and improving the green image. It underscores the proactive role of businesses in engaging with stakeholders to foster social responsibility. By following these guidelines, companies can increase transparency and traceability in their operations, thereby boosting public trust in their environmental initiatives and cultivating a positive social reputation [3, 4].

The financial crisis of 2008 profoundly impacted global stock markets, including China's, leading to prolonged market volatility [5]. In 2020, it was observed that most listed companies, especially in heavily polluting sectors, selectively disclosed negative environmental information, posing substantial risks to long-term corporate sustainability. This selective disclosure, particularly prevalent in heavy-polluting industries, has the potential to exacerbate negative news circulation, causing stock price crashes and market instability. While various studies have examined stock price collapses from different angles, such as investor behavior, analyst predictions, and accounting information quality, research focusing on the impact of EMS certification, particularly in heavily polluting industries, remains limited [6–8].

Stock price collapses have a severe impact on financial markets and hinder the high-quality development of the real economy. The 20th Congress of the Communist Party of China explicitly called for strengthening the financial stability protection system and preventing systemic risks, highlighting the importance of financial stability for the stable development of the economy and society. Heavy polluting industries primarily consist of heavy industrial enterprises, which account for a significant share of China's total industrial output. Conducting further research on the factors influencing the risk of stock price collapse from the perspective of environmental management system certification, especially within the context of heavily polluting industries, will not only help mitigate financial risks but also play a crucial role in promoting the high-quality development of the real economy and maintaining national economic security [9, 10].

While existing academic literature has examined the environmental impact of EMS certification, there is a lack of research on its economic effects. Some research has indicated that EMS certification can improve companies' environmental performance [7, 8, 11]. However, other scholars have found that EMS certification has a limited effect on enhancing a company's environmental performance[12–14]. Existing studies have primarily focused on the innovation effects of certified firms and have found that EMS certification significantly promotes corporate innovation (He & Shen, 2019; Ren et al., 2020) [15, 16]. However, further research is needed to explore the effectiveness of environmental management system certification in the capital market.

This paper examines the impact of environmental management system certification on the risk of stock price collapse and its underlying mechanisms. The findings can provide theoretical support and policy guidance for the government to improve the environmental management system certification system and enhance the stability of the capital market. Additionally, it can help enterprises optimize their strategic decision-making to increase corporate value.

Theoretically, certification of an environmental management system can reduce the risk of a collapse in stock prices by improving both corporate environmental performance and information transparency. On one hand, a company's environmental governance performance is an internal prerequisite that contributes to investment in environmental protection and

management in accordance with the provisions of the environmental management system. Companies that fail to comply with environmental management system standards, such as lacking an environmental responsibility philosophy or exceeding prescribed pollution limits, will be disqualified from obtaining environmental management system certification. After obtaining environmental management system certification, companies must undergo environmental monitoring by an independent third-party certification body. This monitoring requires the company to continuously improve its environmental performance [17].

Another aspect is the information disclosure mechanism adopted by companies. Improving their environmental performance increases the likelihood of disclosing negative environmental information. Disclosure of positive environmental information can sometimes conceal negative environmental impacts, known as the 'masking effect' [18]. In an efficient capital market, if positive environmental information outweighs negative environmental information, it can have a net positive environmental effect on the firm without adversely affecting its share price. Therefore, objective internal management practices require timely disclosure of both positive and negative environmental information. It is important to note that positive environmental information obtained through EMS certification should not be used to mask negative environmental information. This approach helps to avoid the accumulation of negative information. As companies disclose more environmental information, their transparency increases. This reduces the accumulation of negative environmental news and significantly dampens the risk of a sharp fall in the company's share price.

Based on the above analysis, this study utilizes data from listed companies in China's high-polluting industry traded in Shanghai and Shenzhen A-shares during the period 2008 to 2020 as the research sample. Firstly, we investigate the impact of certification of an environmental management system on the risk of a company's stock price crashing; secondly, we analyze the direct effect of corporate governance mechanisms and information disclosure on the connection between environmental management system certification and stock price crash risk. Finally, we investigate the moderating impact of relevant institutional factors of media participation on the certification of environmental management systems and the risk of stock price crashes, focusing on corporate media participation. The findings indicate that the certification of environmental management systems considerably mitigates the risk of a fall in stock prices. Prior to the introduction and implementation of the new Environmental Protection Law in 2015, this effect was not significant, yet grew increasingly pronounced over time. The primary channel through which environmental management system certification deters such risks relates to enhanced transparency regarding corporate information and environmental performance. These results suggest that environmental management system certification might serve as a tool for environmental governance, one capable of curbing the collapse of stock prices. EMS certification can serve as an "environmental governance tool" to prevent the decline of stock prices and improve corporate transparency as an "information transfer tool." The risk of stock price decline is inhibited as a result. Media engagement by corporations has a favorable impact on how EMS certification influences stock price risk. Our results will enhance comprehension of the certification of environmental management systems, augment research content, and furnish empirical proof to enhance the governance system for limiting environmental pollution while regulating corporate conduct in the implementation of environmental management system certification, as well as preventing a decline in the stock price of the company.

The research contribution of this paper lies in three main aspects. Firstly, current empirical studies on the correlation between environmental management system certification and the risk of corporate stock price collapse are predicated on developed country contexts and lack empirical evidence from developing countries. Despite this, China holds the world's leading position concerning the number of firms certified to ISO 14001, yet few empirical studies exist

on this relationship. We address this gap by investigating whether EMS certification can mitigate the risk of a stock price collapse in our setting. This analysis aims to bridge the knowledge gap and enhance comprehension of EMS certification. Additionally, prior studies evaluating the impact of environmental management system certification on mitigating the danger of a stock price collapse have not taken into account the circumstances in which the efficacy of environmental management system certification is achieved. Therefore, considering the fundamental analysis of how obtaining environmental management system certification can mitigate the risk of stock price decline, we investigate the intervening function of corporate environmental performance and external information disclosure in the association between environmental management system certification and corporate environmental performance. This investigation approaches from the perspectives of corporate governance mechanisms and information disclosure mechanisms. This study aims to elucidate the mechanisms by which environmental management system certification affects stock price collapse, offering a fresh perspective on evaluating environmental management system certification efficacy. Thirdly, the empirical analysis examines the risk of environmental management system certification impacting the collapse of a company's stock price. The effectiveness of the certification is tested at varying levels of property rights and marketisation, elucidating the essential role of factors such as enterprise nature and meso-market systems in its effectiveness. This analysis provides empirical evidence supporting the effectiveness of environmental laws and regulations, including the new environmental protection law. The study additionally presents empirical data on the efficacy of implementing the new legislation on environmental protection, as well as other related laws and regulations.

## 2. Literature review

Since the 20th century, stock price crashes have significantly impacted the stability of global financial markets. This instability jeopardizes the development of the real economy and leads to substantial losses for investors. Previous research by foreign scholars has primarily focused on market dimensions to explain stock price crashes, proposing hypotheses such as the financial leverage effect, stock price bubble, and information incompleteness based on theoretical frameworks like information incompleteness and behavioral finance [19, 20]. Kim et al. [19] found that market prices do not fully reflect investors' private information due to factors such as transaction costs and the credibility of information. Ren et al. [20] discovered that selective information disclosure by internal management creates an information gap between the firm and external investors, leading to overvaluation of the share price and formation of an economic bubble, ultimately resulting in a stock price crash.

In contrast, China's capital market, as an emerging market, exhibits significant differences in market stability and maturity compared to Western countries. Zhou et al. [21]verified a positive correlation between investors' heterogeneous beliefs and the risk of stock price collapse, suggesting that analysts' involvement can mitigate information asymmetry between companies and investors [22]. However, analysts' optimistic biases can sometimes increase the risk of stock price collapse for listed companies [23]. While domestic and international literature widely explores the formation mechanisms and influencing factors of stock price collapse risk, research on the relationship between Environmental Management System (EMS) certification and stock price collapse risk is relatively sparse.

The relationship between Environmental Management System (EMS) certification and idiosyncratic risk has garnered significant attention and research. Tzouvanas et al. [24]found that environmental disclosure significantly reduces the idiosyncratic risk of investment in European manufacturing firms, a result that can be explained by stakeholder and legitimacy

theories but not by managerial opportunism. Kong et al. [25] further confirmed that Environmental Corporate Social Responsibility (ECSR) also reduces idiosyncratic risk among A-share listed firms in China, with a more pronounced effect in state-owned firms, firms with weaker external monitoring mechanisms, and firms with poor internal control. Xue et al. [26] examined the multidimensional corporate environmental performance (CEP) and found that Environmental Management Performance (EMP) effectively reduces firm risk, especially in the manufacturing sector, while Environmental Operational Performance (EOP) does not have a significant impact on firm risk. Additionally, Zhang et al. [27] studied the impact of EMS certification on firms' access to finance and found that the certification helps alleviate financial capital constraints by gaining stakeholder approval and support, particularly when the government emphasizes environmental protection and the company has a good environmental record and financial position. Finally, Arimura et al. [28] noted that the effectiveness of ISO 14001 in reducing pollution varies across countries and environmental impacts, being more effective when there are clear cost-saving opportunities. In summary, EMS certification plays a crucial role in reducing idiosyncratic risk and improving firms' access to finance, but its specific effects depend on various factors, including national environmental policies and the company's environmental management level.

The impact of environmental management system certification on stock price crash risk necessitates further comprehensive exploration. This paper aims to fill this gap by examining the impact of EMS certification on the risk of stock price collapse and its underlying mechanisms. The research contributions of this paper can be summarized into three main points: (1) This study provides empirical evidence from developing countries, particularly China, which holds the world's leading position concerning the number of firms certified to ISO 14001. (2) The study investigates the intervening function of corporate environmental performance and external information disclosure in the association between environmental management system certification and stock price crash risk. (3) The effectiveness of EMS certification is tested at varying levels of property rights and marketisation, elucidating the essential role of factors such as enterprise nature and meso-market systems in its effectiveness.

By addressing these aspects, this paper aims to offer theoretical support and empirical analysis to help the government and relevant institutions understand the influence of EMS certification, thereby enhancing the stability of the capital market and promoting high-quality economic development.

## 3. Theoretical analysis and research hypothesis

### 3.1. Environmental management system certification and the risk of stock price collapse

Research on the factors related to the risk of stock price collapse can be summarized into two main categories: principal-agent theory and information asymmetry theory. The principal-agent theory suggests that corporate executives have a tendency to hide negative news for their own benefit, such as stock price increases. When negative news continues to accumulate, it may lead to a stock price collapse [9]. The information asymmetry theory posits that there is significant asymmetry in the degree of negative information grasped by internal and external investors. If negative news accumulates over a long period and is suddenly revealed to external investors, there will be a massive sell-off and a stock price crash. Lin and Wu [10] point out that the concealment of environmental information is more evident than other types of idiosyncratic information, so the accumulation of such negative information may place the firm at greater risk of a stock price collapse.

The certification of environmental management systems (EMS) may have the potential to mitigate the risk of a stock price collapse through governance and information effects. The governance effect pertains to the obligation of companies to publicly disclose their environmental management practices subsequent to obtaining certification for their environmental management system. This disclosure is overseen by a third-party organization to ensure adherence to environmental regulations across the production chain, including raw material sourcing and production processes. Enhancing the company's environmental performance can improve environmental compliance, reduce negative environmental news, and potentially mitigate the risk of a decline in the company's stock price. EMS certification has been shown to positively contribute to improving the environmental performance and compliance of enterprises [29].

The information effect refers to the positive impact of EMS certification in promoting the disclosure of environmental information by companies and increasing information transparency, which in turn reduces the risk of stock price collapse. EMS certification has led to a gradual improvement in the environmental performance of enterprises, and companies with good environmental performance are more willing to disclose environmental information to win societal goodwill [30]. Furthermore, companies can use the positive effects of EMS certification to offset the negative effects of negative environmental information. When a company's environmental information transparency continues to improve, the accumulation and outbreak of negative news are less likely, effectively controlling the risk of stock price collapse [31].

Therefore, EMS is a voluntary and participatory environmental regulation that aims to enhance corporate environmental compliance and promote timely disclosure of information. This can lead to improved environmental performance and greater transparency of information, reducing the likelihood of negative news accumulation and outbreaks, and effectively managing the risk of stock price collapse. Based on this analysis, the following hypothesis is proposed:

H1: Environmental management system certification has a negative effect on the risk of stock price collapse.

## 3.2. Nature of property rights, environmental management system certification, and the risk of stock price collapse

The development of enterprises in China is influenced by various factors, including national policies and the nature of the enterprise, such as whether it is state-owned or privately owned [32]. These factors play a crucial role in shaping the enterprise's development trajectory. It is expected that the inhibitory effect of EMS certification on the risk of stock price collapse will be more significant in state-owned enterprises (SOEs) compared to non-state-owned enterprises.

Firstly, SOEs are more comprehensively regulated by the government compared to non-SOEs. They have a stronger awareness and capability to fulfill their environmental duties [33]. Secondly, managers within SOEs usually exhibit stronger regulatory motivation in environmental management compared to their counterparts in non-SOEs(Tang,2020) [34]. Thirdly, SOEs generally have a stronger hierarchical mindset and more executive power to intervene in corporate social responsibility (CSR) through administrative means, thereby fulfilling their environmental responsibilities [35].

The theory of property rights provides a foundation for understanding these differences. Property rights influence environmental management by aligning incentives for sustainability

and economic prosperity. For example, assigning property rights to natural resources can lead to more effective environmental management and reduce the risk of resource depletion and associated financial risks [36]. In the context of fisheries, property rights such as individual transferable quotas (ITQs) or territorial use rights can enhance sustainability [37]. Moreover, research suggests that property rights regimes can impact the environmental performance of enterprises. In New Zealand, the management of fisheries resources under a rights-based framework demonstrated how property rights could contribute to better environmental outcomes and economic stability [38]. Similarly, allowing for "nonuse rights" in public natural resources can advance environmental goals and lead to more stable economic outcomes by reducing contention over resource use [39].

A comparative study of Chinese multinationals revealed that non-state-owned enterprises (non-SOEs) outperform state-owned enterprises (SOEs) in terms of environmental and governance performance. This finding suggests that non-SOEs may be more efficient in integrating environmental management practices due to different incentive structures and operational flexibility [40]. Conversely, central SOEs in China have been found to significantly violate environmental regulations, leading to numerous pollution incidents [41]. This indicates potential compliance challenges within SOEs despite stringent regulations.

Furthermore, the propensity for environmental legitimacy pressure differs between SOEs and non-SOEs. Non-SOEs are more strongly associated with impression management of carbon information disclosure, reflecting a greater responsiveness to environmental legitimacy pressures compared to SOEs [42]. Additionally, the impact of coercive and non-coercive environmental supply chain sustainability drivers varies, with SOEs potentially more influenced by coercive drivers due to their regulatory environment [43].

Therefore, this paper introduces the variable of the nature of property rights and proposes the following hypothesis:

H2: The inhibitory effect of environmental management system certification on the risk of company stock price collapse is more pronounced in state-owned enterprises.

## 3.3. Marketization level, environmental management system certification, and the risk of stock price collapse

The impact of marketization on the risk of a share price crash can vary significantly between industries due to factors such as technology and resources [44]. Consequently, the level of marketization varies considerably across industries. Insufficient marketization may lead to inadequate institutional and legal safeguards in the business environment, resulting in ineffective constraints on the environmental performance of enterprises [27]. As a result, the impact of environmental management system (EMS) certification on the volatility of firms' stock prices may be relatively small [45], leading to an insignificant dampening effect on the risk of stock price crashes from an external perspective [28].

However, when the level of marketization is higher, firms are subject to greater external supervision, which increases their awareness and efforts to obtain EMS certification. Stricter market supervision will have a more significant inhibiting effect on the risk of stock price collapse.

The impact of marketization on the risk of a share price crash can vary significantly between industries due to factors such as technology and resources [44]. Consequently, the level of marketization varies considerably across industries. Insufficient marketization may lead to inadequate institutional and legal safeguards in the business environment, resulting in ineffective constraints on the environmental performance of enterprises [27]. As a result, the

impact of environmental management system (EMS) certification on the volatility of firms' stock prices may be relatively small [45], leading to an insignificant dampening effect on the risk of stock price crashes from an external perspective [28].

However, when the level of marketization is higher, firms are subject to greater external supervision, which increases their awareness and efforts to obtain EMS certification. Stricter market supervision will have a more significant inhibiting effect on the risk of stock price collapse.

From a theoretical perspective, the principal-agent theory and stakeholder theory provide a solid foundation for understanding the relationship between marketization, EMS certification, and stock price collapse risk. According to principal-agent theory, the separation of ownership and control in corporations can lead to conflicts of interest between managers and shareholders. Higher levels of marketization enhance market mechanisms and external monitoring, reducing information asymmetry and aligning the interests of managers and shareholders, thus improving corporate governance and performance (Jensen & Meckling, 1976) [46]. Stakeholder theory emphasizes the importance of various stakeholders, including investors, regulators, and the community, in influencing corporate behavior [47]. In highly marketized regions, firms face greater pressure from stakeholders to adopt sustainable practices and disclose environmental performance. This pressure incentivizes firms to obtain EMS certification, thereby reducing environmental risks and enhancing transparency, which in turn lowers the risk of stock price crashes.

Empirical studies have shown that the Carbon Emission Trading System (CETS) in China significantly reduces the risk of stock price crashes for heavily polluting companies, particularly in pilot areas, by reducing the skewness and volatility of firm-specific returns [48]. This suggests that enhanced market mechanisms and regulatory environments can amplify the benefits of EMS certification. Additionally, improving internal controls and information disclosure quality has been demonstrated to mitigate the risk of future stock price collapses [49]. Higher ESG scores are correlated with reduced stock price collapse risk, highlighting the protective role of ESG factors in stabilizing stock prices [50]. This supports the notion that in more marketized environments, where ESG practices are better integrated and monitored, the impact of EMS certification on reducing stock price crash risk is more pronounced.

Based on the above analysis, this paper introduces the degree of marketization variable and proposes the following hypothesis:

H3: Firms in regions with a higher degree of marketisation will experience a more significant reduction in share price crash risk with EMS certification.

The paper's theoretical framework is illustrated in (Fig 1).

# 4. Research design

## 4.1. Formatting of mathematical components

The data sample used in this paper consists of A-share listed companies with heavy pollution from 2008 to 2020. Heavy polluting enterprises are defined based on the Guide to Environmental Information Disclosure for Listed Companies [7], which includes 16 types of industries: thermal power, iron and steel, cement, electrolytic aluminum, coal, metallurgy, chemical, petrochemical, building materials, paper, brewing, pharmaceutical, fermentation, textile, tannery, and mining. During the sample selection process, certain criteria were applied. Data from companies that were delisted or had been listed for less than one year were excluded. Additionally, data with significant outliers and incomplete information for some indicators were also excluded. After applying these screening criteria, a final sample of 469 companies

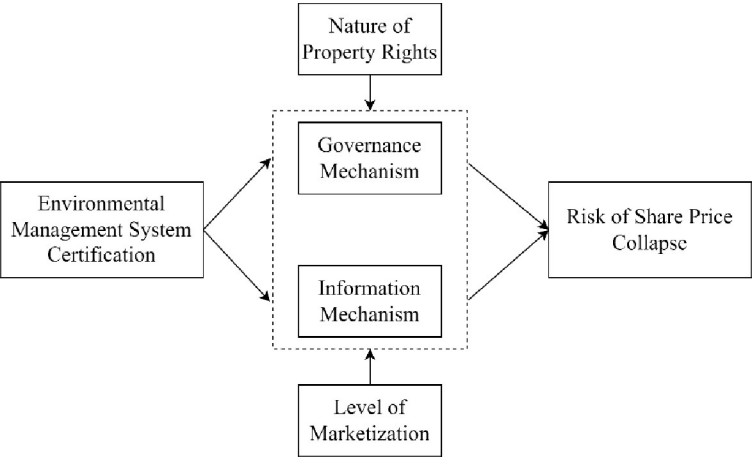

**Fig 1. Theoretical framework.**

with a total of 5,794 data samples was obtained. To address any missing sample data, the variables underwent a 1% trim on both ends of the distribution to reduce the impact of outliers before and after the trimming process.

## 4.2. Variable definition

In this paper, the risk of stock price collapse is used as the explained variable. The research method follows the approach used by Chen, J., Hong, H., & Stein, J. C. [51], where down-to-up volatility (DUV) and negative coefficient of skewness (NCSK) are selected as indicators of stock price crash risk. These indicators are calculated as shown in Eqs (1) and (2).

$$\mathrm{DUV}_{i,t} = \ln\{[(n_{i,t}-1)\sum\nolimits_{down}R_d^2]/[n_d-1]\sum\nolimits_{up}R_d^2\} \tag{1}$$

$$\mathrm{NCSK}_{i,t} = -\left[n(n-1)^{\frac{3}{2}}\sum W_{i,t}^3\right]/[(n-1)(n-2)(\sum W_{i,t}^2)^{3/2}] \tag{2}$$

Where i is the firm, t is the year, EMSC is Environmental Management System Certification, W is the market-adjusted weekly return, n is the number of weeks the stock trades per year, Ind is the number of weeks with weekly returns > the average annual return, and nu is the number of weeks with weekly returns less than the average annual return.

The explanatory variable in this study is environmental management system certification (EMSC). It is measured using a dummy variable approach, where a value of 1 indicates that the company has completed environmental management system certification, and a value of 0 indicates otherwise. The certification status is assessed based on the requirement that the certification is completed within a natural year for a period of at least six months.

Referring to the established research [19], the firm size (Size), net asset-to-book ratio (MB), information transparency (Abs), gearing ratio (Lev), return on assets (ROA), and management shareholding ratio (Msh), are selected as control variables in this paper. The specific variables are defined as shown in Table 1.

## 4.3. Model construction

Based on the existing literature [6, 52], and the purpose of this paper, the following empirical model was constructed to test the correlation effect of environmental management system

**Table 1. Description of the main variables in this paper.**

| Variable Type | Variable name and symbol | Variable Explanation |
|---|---|---|
| Explained variables | Stock Return Volatility (DUV) | Calculated as Eq (1) |
| | Negative return skewness coefficient (NCSK) | Calculated as Eq (2) |
| Explanatory variable | Environmental Management System Certification (EMSC) | The authentication count is 1, and the opposite is 0 |
| Mediator variables | Corporate Environmental Performance (SYN) | Environmental tax on business income of 10,000 yuan unit and add 1 to take the logarithm |
| | Corporate Information Transparency (OPA) | Information disclosure evaluation indexes constructed from five aspects of constructing disclosure vehicles, environmental management, environmental liabilities, environmental regulation and certification, and environmental performance and governance, with quantitative disclosure as 2, qualitative disclosure as 1, and no disclosure as 0 |
| Moderator variable | Media attention (RSS) | Number of media reports and add 1 to take the natural logarithm |
| Grouping variables | Nature of Ownership (SOE) | Defined as 1 if state-owned, 0 otherwise |
| | Level of marketisation (HHI) | The square of the ratio of the total ownership interest of each company in the industry to the total ownership interest in the industry is cumulated |
| Control variables | Enterprise size (Size) | Take the natural logarithm of the total assets |
| | Average weekly return (Re) | Annual average of weekly returns |
| | Information Transparency (Abs) | Absolute value of controllable profit to be counted by the enterprise |
| | Gearing ratio (Lev) | Total liabilities as a percentage of total asset value |
| | Return on Assets (ROA) | Total profit as a percentage of total assets |
| | Management shareholding ratio (Msh) | Number of shares held by management to total shares |
| | Year (YEAR) | Year of affiliation |
| | Industry (IND) | Industry |

certification on the risk of stock price collapse and the increase of institutional holdings.

$$NCSK_{i,t+1} = \alpha_0 + \beta_1 EMSC_{i,t} + \sum Controls_{i,t} + \sum Ind + \sum Year + \varepsilon_{i,t} \quad (3)$$

$$DUV_{i,t+1} = \alpha_0 + \alpha_1 EMSC_{i,t} + \sum Controls_{i,t} + \sum Ind + \sum Year + \varepsilon_{i,t} \quad (4)$$

Where $\sum Controls_{i,t}$ are control variables, the $\sum Year$ and $\sum Ind$ are annual dummy variables and industry dummy variables, respectively. $\alpha_0$、 $\beta_1$、 $\alpha_1$ are constant terms, and $\varepsilon$ are the random error numbers.

## 5. Empirical results and analysis

### 5.1. Descriptive statistics

The results of descriptive statistics for the main variables are shown in Table 2. The maximum value of stock return volatility (DUV) is 2.417, the minimum value is -2.154, and the mean value is -0.214, and the maximum and minimum values of NCSK are 4.002 and -3.857, respectively, with a mean value of 0.751. It can be found that both stock return volatility and negative return skewness coefficients are approximately consistent with the theoretically predetermined normal distribution model, and there are more obvious individual differences. The mean value of environmental management system certification (EMSC) of the companies involved in this study is 0.398%, and it can be seen that the percentage of environmental management system certification in the sample is about 39.8%, which still has a large gap compared to the known levels in developed countries in Europe and America.

**Table 2. Descriptive statistics results.**

| Variables | N(sample size) | Average value | Maximum value | Minimum value | Standard deviation |
|---|---|---|---|---|---|
| DUV | 5,794 | -0.214 | 2.417 | -2.154 | 0.429 |
| NCSK | 5,794 | -0.287 | 4.002 | -3.857 | 0.751 |
| EMSC | 5,794 | 0.398 | 1 | 0 | 0.365 |
| Size | 5,794 | 21.419 | 25.810 | 17.746 | 1.943 |
| Re | 5,794 | 0.005 | 0.032 | -0.019 | 0.009 |
| Abs | 5,794 | 0.072 | 0.295 | 0.003 | 0.058 |
| Lev | 5,794 | 0.463 | 0.894 | 0.062 | 0.217 |
| Roa | 5,794 | 0.037 | 0.241 | -0.135 | 0.230 |
| Msh | 5,794 | 0.102 | 0.92 | 0 | 0.125 |

## 5.2. Correlation analysis

The mean and standard deviation and the matrix of correlation coefficients between the variables studied in this paper are shown in Table 3. As shown in the results of Table 3, the environmental management system certification (EMSCt) is significantly negatively correlated with the volatility of stock returns (DUV t+1) and the negative coefficient of skewness of returns (NCSKt+1) ($\beta 1$ = -0.405, P<0.001; $\beta 2$ = -0.236, P<0.001) each of the correlation coefficient matrices between the main variables were all below 0.5, indicating significant correlation and excluding the problem of multicollinearity. The above results provide initial support for the preliminary hypothesis testing.

## 5.3. Benchmark analysis

The regression results in Table 4 show that the regression coefficients between environmental management system certification (EMSCt) and stock return volatility (DUVt+1) and negative return skewness coefficient (NCSK t+1) are -0.084 and -0.097, respectively, and the results are significantly different at the 1% confidence level.

Certification of companies with environmental management systems is effective in avoiding the risk of stock price collapse. On average, the stock return volatility of certified companies decreases by 0.084 units and the negative return skewness coefficient decreases by 0.097 units compared to those listed companies without environmental management system

**Table 3. Correlation analysis.**

| Variables | 1 | 2 | 3 | 4 | 5 | 6 | 7 | 8 |
|---|---|---|---|---|---|---|---|---|
| 1 EMSC | 1 | | | | | | | |
| 2 NCSK | -0.405*** | 1 | | | | | | |
| 3 DUV | -0.236** | 0.297** | 1 | | | | | |
| 4 Size | 0.269*** | 0.361*** | 0.299*** | 1 | | | | |
| 5 Ret | 0.356** | 0.308** | 0.290** | 0.282*** | 1 | | | |
| 6 Abs | 0.454** | 0.394*** | 0.514** | 0.236* | 0.248*** | 1 | | |
| 7 Lev | 0.627*** | 0.418*** | 0.215*** | 0.382*** | -0.102*** | 0.286*** | 1 | |
| 8 ROA | -0.154*** | -0.039** | 0.036** | -0.290*** | 0.087*** | 0.261*** | 0.211*** | 1 |

Note

*** p < 0.01

** p < 0.05

* p < 0.1

**Table 4. Benchmark regression results.**

| Variables | DUV $_{t+1}$ | NCSK$_{t+1}$ |
|---|---|---|
| EMSC $_t$ | -0.084** | -0.097** |
| Size $_t$ | 0.014 | 0.029* |
| Re $_t$ | 9.416* | 13.285* |
| Abs $_t$ | 0.276** | 0.348** |
| Lev $_t$ | 0.054 | 0.062 |
| Roa $_t$ | -0.084** | -0.097** |
| Year | Control | Control |
| Ind | Control | Control |
| Adjusted R$^2$ | 0.056 | 0.062 |
| N | 5,794 | 5,794 |

Note

*** p < 0.01

** p < 0.05

* p < 0.1

certification. This is consistent with the hypothesis of existing studies, and further the hypothesis H1 of this paper is verified.

## 6. Further analysis

### 6.1. Intermediary effect mechanism

Models (5) and (6) were constructed based on theoretical analysis to test the influence mechanism of environmental management system certification in suppressing the risk of stock price collapse of a company. The models aim to improve the company's environmental performance and promote the company's information disclosure.

$$MV_t = \alpha_0 + \beta_1 EMSC_t + \beta_2 Controls_t + \sum Year + \sum Ind + \varepsilon_{i,t} \tag{5}$$

$$NCS_{t+1}/DUV_{t+1} = \alpha_0 + \beta_1 EMSC_t + \beta_3 MV_t + \beta_4 Controls_t + \sum Year + \sum Ind + \varepsilon_{i,t} \tag{6}$$

MVt is the mediating variable, including corporate environmental performance (SYN) and corporate information transparency (OPA). The measurement of corporate environmental performance draws on the research method of Jiang et al. [17], using the environmental tax of 10,000 yuan unit business income (called "sewage fee" before 2018) as a proxy for corporate environmental performance, and +1 taking the natural logarithm to make it more in line with the assumption of normal distribution. The higher the amount of environmental taxes per unit of operating income in 10,000 yuan, the worse the firm's environmental performance. The measurement of corporate information transparency draws on the research method of Ren et al. [16] to construct an environmental information disclosure evaluation index system from five major aspects: disclosure carriers, environmental management, environmental liabilities, environmental regulation and certification, and environmental performance and governance, assigning a value of 0 when there is no disclosure, 1 when qualitative disclosure is made, and 2 when quantitative disclosure is made.

Table 5 shows the test results (1) and (2), indicating a significant positive impact of environmental management system certification on the company's environmental performance and information transparency. After taking control variables into account, the impact coefficients

**Table 5. Test of mediation mechanism between environmental management system certification and stock crash risk results.**

| Variables | $SYN_t$ | $OPA_t$ | $DUV_{t+1}$ | | $NCSK_{t+1}$ | |
|---|---|---|---|---|---|---|
| | (1) | (2) | (3) | (4) | (5) | (6) |
| $EMSC_t$ | -0.065** | 0.047** | -0.342* | -0.217* | -0.143* | -0.157* |
| $SYN_t$ | | | 0.037* | | 0.062** | / |
| $OPA_t$ | | | | -0.042* | | -0.054* |
| $Size_t$ | 0.009* | 0.015 | 0.731* | 0.431 | 0.732 | 0.519 |
| $Re_t$ | 5.431* | 3.749 | 1.430 | 1.829* | 2.641 | 1.526* |
| $Abs_t$ | 0.162** | 0.085** | 0.284** | 0.316** | 0.097 | 0.193 |
| $Lev_t$ | 0.416 | 0.754 | 0.483 | 0.372 | 0.519 | 0.531* |
| $Roa_t$ | -0.721** | -0.354* | -0.5432 | -0.179 | -0.238 | -0.653 |
| Year | Control | Control | Control | Control | Control | Control |
| Ind | Control | Control | Control | Control | Control | Control |
| Adjusted $R^2$ | 0.075 | 0.089 | 0.064 | 0.062 | 0.67 | 0.72 |

Note

*** $p < 0.01$

** $p < 0.05$

* $p < 0.1$

are -0.065 and 0.047, respectively, passing the significance test of 5%. This suggests that environmental management system certification effectively promotes the company's environmental performance, accelerates the enterprise's construction of environmental legitimacy and compliance, and improves its environmental performance. The results (3) and (5) indicate that the company's environmental performance significantly inhibits the risk of stock price crash (= 0.037, P<0.1; = 0.062, P<0.1). Similarly, the results (4) and (6) show that the transparency of the company's information also significantly inhibits the risk of stock price crash (= -0. The results ($\beta$ = -0.042, P<0.1; $\beta$ = -0.054, P<0.1) suggest that enhancing information transparency can effectively enhance companies' environmental performance, accelerate the establishment of environmental legality and compliance, and improve their environmental performance. The analysis indicates that improving information transparency can effectively inhibit the risk of stock price collapse. Improving the company's environmental performance and information transparency in the environmental management system certification plays an intermediary role in reducing the risk of stock price collapse. There is an environmental management system certification that aims to improve the company's environmental performance. This mechanism also helps to suppress the risk of stock price collapse by improving information transparency. The theoretical analyses presented in the previous part of this paper are verified by this certification. Environmental management system certification suppresses the risk of stock price collapse by improving corporate environmental performance and information transparency.

## 6.2. Robustness test

To test the robustness of the results in this paper, the results are further tested for robustness using the substitution variable method, double difference method, and propensity score matching method.

**6.2.1. Substitution variable method test.** Considering that the negative return bias coefficient and stock return volatility have certain variable bias in market development and research, which leads to the error problem of stock price crash risk measurement with this benchmark;

**Table 6. Regression results for the test of alternative variables.**

| Variables | DUF $_{t+1}$ | INC $_{t+1}$ | INR$_{t+1}$ |
|---|---|---|---|
| EMSC $_t$ | -0.143** | -0.027** | -0.093** |
| Size $_t$ | -0.032 | 0.641 | 0.265 |
| Re $_t$ | 2.316 | 4.179 | 3.853 |
| Abs $_t$ | 0.391 | 0.094** | 0.337** |
| Lev $_t$ | 0.047 | 0.056 | 0.021 |
| Roa $_t$ | -0.052 | -0.004** | -0.016** |
| Year | Control | Control | Control |
| Ind | Control | Control | Control |
| Adjusted R$^2$ | 0.012 | 0.030 | 0.005 |

Note

*** $p < 0.01$

** $p < 0.05$

* $p < 0.1$

at the same time, considering the existence of certain subjectivity of institutional shareholding ratio, which in turn affects the scientific and generalizability of the research results. Therefore, in the robustness test, this paper selects the difference between the rising frequency and falling frequency of stock returns (DUF, INC and INR) as a proxy for stock price crash risk, and the final results show that the mechanism between environmental management system certification and stock return volatility is significantly negative, indicating that the previous conclusions still hold after the proxy variable test. The specific results are shown in Table 6.

**6.2.2. Difference in difference method test.** To address potential endogeneity issues, the paper employs the difference in difference method for robustness testing. Following Wing et al. [53], a double-difference model is constructed using the midpoint of the first completion of environmental management system certification. The interaction term between the group dummy and the time dummy (Du×Dt) is defined. If an enterprise meets the inclusion criteria of the experimental group and is in the year of the first certification, the value is 1; otherwise, it is 0. The regression results of the double-difference method are presented in Table 7, showing

**Table 7. Regression results of double difference method test.**

| Variables | DUV $_{t+1}$ | NCSK$_{t+1}$ |
|---|---|---|
| Du$_t$ × Dt$_t$ | -0.213** | -0.184** |
| Size $_t$ | 0.016 | 0.072** |
| Re $_t$ | 7.648** | 12.354** |
| Abs $_t$ | 0.357* | 0.386* |
| Lev $_t$ | -0.075 | -0.136 |
| Roa $_t$ | -0.184 | -0.375 |
| Firm | Control | Control |
| Year | Control | Control |
| Ind | No control | No control |
| Adjusted R$^2$ | 0.042 | 0.013 |

Note: *** $p < 0.01$

** $p < 0.05$

* $p < 0.1$

**Table 8. Balance test results.**

| Variables | Before matching | | | After matching | | |
|---|---|---|---|---|---|---|
| | Control group | Experimental group | Differences | Control group | Experimental group | Differences |
| Size | 25.756 | 25.802 | 0.046 | 25.806 | 25.802 | -0.004** |
| Lev | 0.517 | 0.498 | -0.019 | 0.493 | 0.498 | 0.005** |
| Roa | 0.060 | 0.072 | 0.012 | 0.074 | 0.072 | -0.002** |
| Dir | 3.154 | 3.146 | -0.008 | 3.145 | 3.146 | 0.001*** |
| Id | 0.242 | 0.237 | -0.005 | 0.237 | 0.237 | 0.000*** |

that the regression coefficients of Du×Dt and DUV and NCSK are negative, consistent with the findings of this study ($\beta = -0.213$, $P<0.05$; $\beta = -0184$, $P<0.05$).

**6.2.3. Propensity score matching method test.** To eliminate systematic differences between the experimental and control groups, the paper conducts a test using the propensity score matching method, as referenced in existing literature [54, 55]. The parameters are set as follows: nearest neighbor matching, matching ratio of 1:1, and Size, Lev, Roa, board size (Dir), and proportion of independent directors (Id) selected as matching covariates [56]. The equilibrium results before and after propensity score matching are shown in Table 8, and the differences represent the values obtained from the experimental group minus the control group. Table 9 displays the regression results of the propensity score matching method, confirming that the regression coefficients of EMSC and DUV and NCSK are negative. This indicates that environmental management system certification is beneficial in reducing the risk of stock price collapse, further supporting the conclusions of the paper.

## 6.3. Test for moderating effects

The extent to which EMS certification has a dampening effect on the risk of stock price collapse is closely related to corporate information transparency. Among them, media involvement is the main factor affecting information transparency [20]. The more the media involvement, the more the public knows about the environmental management system

**Table 9. Regression results of propensity score matching method test.**

| Variables | DUV $_{t+1}$ | NCSK$_{t+1}$ |
|---|---|---|
| DUV $_t$ | 0.037** | 1 |
| NCSK $_t$ | 1 | 0.059** |
| INI_buy $_t$ | -0.025** | -0.031** |
| EMSC $_t$ | -0.087** | -0.042** |
| Size $_t$ | 0.017** | 0.009* |
| Re $_t$ | 12.554** | 11.814** |
| Abs $_t$ | 0.513* | 0.462* |
| Lev $_t$ | 0.042 | 0.031 |
| Roa $_t$ | -0.025** | -0.049** |
| Firm | No control | No control |
| Year | Control | Control |
| Ind | Control | Control |
| Adjusted R$^2$ | 0.036 | 0.012 |

Note

* and ** and ***indicate significant at 10% and 5% and 1% confidence level, respectively.

**Table 10. Moderating effects results.**

| Variables | $DUV_{t+1}$ | $NCSK_{t+1}$ |
|---|---|---|
| $EMSC_t$ | -0.183** | -0.902** |
| RSS | -1.254* | -1.373* |
| $EMSC_t$* RSS | -0.765*** | -0.287*** |
| $Size_t$ | -0.236* | -0.021 |
| $Re_t$ | 13.529** | 6.294** |
| $Abs_t$ | 2.185** | 1.854** |
| $Lev_t$ | 0.754 | 0.643 |
| $Roa_t$ | -0.095** | -0.163** |
| Year | Control | Control |
| Ind | Control | Control |
| Adjusted $R^2$ | 0.003 | 0.005 |

certification, and the more credible the suppression effect on the stock price crash risk. Therefore, this paper selects media involvement (RSS) as a moderating variable to investigate the moderating effect of media involvement on the mechanism between environmental system certification and stock price crash. According to the results in Table 10, it can be seen that the coefficient of the interaction term between environmental management system certification and media involvement in columns (1) and (2) is significantly negative at the 1% level, indicating that improving the market environment at the level of information transparency does inhibit the increased risk of stock price collapse. There is a more significant governance effect.

## 6.4. Heterogeneity analysis

**6.4.1. Heterogeneity analysis based on the nature of property rights.** To investigate the mechanism relationship between different property rights nature on environmental management system certification and stock price crash risk, this paper conducts a heterogeneity test of relevant property rights nature. Table 11 shows that environmental management system certification significantly inhibits stock price collapse in state-owned enterprises, while the inhibitory effect on stock price collapse risk is not significant in non-state-owned enterprises, so if the enterprise belongs to state-owned enterprises, the more significant the inhibitory effect of environmental management system certification on stock price collapse risk is, which verifies hypothesis H2.

**Table 11. Environmental management system certification and share price collapse risk: Differentiating between corporate systems.**

| Variables | State-owned enterprises | | Non-State Owned Enterprises | |
|---|---|---|---|---|
| | $DUV_{t+1}$ | $NCSK_{t+1}$ | $DUV_{t+1}$ | $NCSK_{t+1}$ |
| $EMSC_t$ | -5.843** | -3.865** | -1.943 | -0.483 |
| $Size_t$ | -0.085** | -0.079** | -0.014* | 0.029* |
| $Re_t$ | 12.435* | 16.285* | 9.837* | 11.329* |
| $Abs_t$ | 0.271** | 0.365** | 0.320** | 0.581** |
| $Lev_t$ | 0.095 | 0.071 | 0.068 | 0.041 |
| $ROA_t$ | -0.064** | -0.038** | -0.142** | -0.085** |
| Year | Control | Control | Control | Control |
| Ind | Control | Control | Control | Control |
| Adjusted $R^2$ | 0.084 | 0.015 | 0.038 | 0.029 |

**Table 12. Environmental management system certification and share price collapse risk: Differentiating market environments.**

| Variables | High level of marketability | | Low level of marketization | |
|---|---|---|---|---|
| | DUV $_{t+1}$ | NCSK$_{t+1}$ | DUV $_{t+1}$ | NCSK$_{t+1}$ |
| EMSC $_t$ | -3.854** | -4.715** | -2.176 | -1.893 |
| Size $_t$ | -0.527* | -0.327* | -0.184 | -0.192 |
| Re $_t$ | 14.596* | 11.954* | 7.543* | 5.492* |
| Abs $_t$ | 0.185** | 0.573** | 0.097** | 0.135** |
| Lev $_t$ | 0.298 | 0.381 | 0.154 | 0.093 |
| Roa $_t$ | -0.495** | -0.362** | -0.186** | -0.127** |
| YEAR | Control | Control | Control | Control |
| Ind | Control | Control | Control | Control |
| Adjusted R$^2$ | 0.011 | 0.021 | 0.032 | 0.073 |

**6.4.2. Heterogeneity analysis based on the level of market environment.** In order to investigate the mechanism relationship between environmental management system certification and stock price collapse risk in different market environments, this paper conducts a heterogeneity test of relevant market environments [56]. Table 12 shows that the correlation coefficients of environmental management system certification are significantly higher in companies in higher marketization levels than in companies in lower marketization levels, and the coefficients all pass the test of heterogeneity, so the higher the marketization level, the more obvious is the inhibitory effect of environmental management system certification on stock price crash risk, which verifies hypothesis H3.

# 7. Conclusion and implications

The study explores the impact of Environmental Management System (EMS) certification on the risk of stock price collapse among heavily polluting enterprises in China, focusing on governance and information disclosure mechanisms. Companies with EMS certification exhibit lower stock volatility and reduced risk of stock price collapse, particularly after the 2015 Environmental Protection Law. This highlights the role of policy changes in enhancing EMS certification's impact, as noted by Boiral and King, Lenox, & Terlaak [57, 58].

The mitigating effect of EMS certification is especially significant for state-owned and heavily polluting enterprises, particularly in regions with high marketization, an aspect that has not been thoroughly explored in previous research. The study also identifies media coverage as a mediating factor, influencing the relationship between EMS certification and stock price collapse risk, building on the findings of Ge et al. [59]. This demonstrates the role of environmental information disclosure in shaping stock market dynamics.

Moreover, EMS certification enhances market trading information and reduces stock price collapse risk by improving a company's environmental performance and transparency. This combined approach provides a comprehensive view beyond the findings of Hossain et al. [60] on corporate environmental commitment and market reaction. The study aligns with Ye et al. [61], who observed that while EMS certification might reduce financial risk, it can also hinder sales growth. This study further explores these impacts under varying degrees of marketization. Similarly, Delmas et al. [62] noted that improved environmental performance might lead to long-term value increases, which this study validates with empirical data.

Despite these contributions, the study has limitations. The focus on heavily polluting enterprises in China limits the generalizability of the findings to other sectors or countries with different regulatory contexts. The analysis primarily covers the period after the 2015 Environmental Protection Law, potentially missing long-term effects or earlier trends.

Additionally, while the study acknowledges the mediating role of media coverage, it does not deeply explore variations across different media types or regions.

These findings have important implications for both government and corporate practices. Governments are encouraged to support EMS certification through policies, regulatory mechanisms, and incentives that promote public awareness of environmental protection, as emphasized by Frondel et al. [63]. Certified facilities tend to exhibit significantly higher financial performance compared to those with only EMS adoption. Enhancing marketization can also promote corporate competition and innovation, thereby improving stock price stability and market trust, which aligns with the findings of Pavlishchuk et al. [64].

For businesses, the study suggests leveraging EMS certification to transition towards environmentally friendly operations and enhance transparency in environmental information, reducing the risk of stock price collapse. Companies should utilize media effectively to improve transparency and governance of stock price risks, reinforcing the role of media highlighted by Ge et al. [59]. Overall, the study not only confirms existing research but also offers new insights into the mechanisms and policy impacts related to EMS certification and stock market behavior.

## Supporting information

**S1 Dataset.**
(XLSX)

## Author Contributions

**Conceptualization:** Hongyu Liu, Pengliang Qiao.

**Data curation:** Hongyu Liu, Pengliang Qiao.

**Formal analysis:** Hongyu Liu.

**Funding acquisition:** Pengliang Qiao.

**Investigation:** Hongyu Liu.

**Methodology:** Hongyu Liu.

**Project administration:** Qin Binbin.

**Resources:** Hongyu Liu.

**Software:** Hongyu Liu, Pengliang Qiao.

**Supervision:** Qin Binbin.

**Validation:** Pengliang Qiao.

**Visualization:** Qin Binbin.

**Writing – original draft:** Hongyu Liu.

**Writing – review & editing:** Qin Binbin.

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
