## [Decision Letter · Decision Letter 0]

24 May 2024

PONE-D-24-16029Does Environmental Management System Certification Keep Enterprises Out of Trouble? Evidence From Stock Price Crash RiskPLOS ONE

Dear Dr. Qiao,

Thank you for submitting your manuscript to PLOS ONE. After careful consideration, we feel that it has merit but does not fully meet PLOS ONE’s publication criteria as it currently stands. Therefore, we invite you to submit a revised version of the manuscript that addresses the points raised during the review process.

We look forward to receiving your revised manuscript.

Kind regards,

Khanh Hoang, Ph.D.

Academic Editor

PLOS ONE

Journal Requirements:

   "This research was funded by National Natural Science Foundation of China (NSFC No. 71962005), Innovation Team Project for Ordinary Universities in Guangdong Province (Humanities and Social Sciences) (2023WCXTD026) and Major Scientific Research Projects in Colleges and Universities in Guangdong (Grant No. 2021ZDJS122)(Grant No. 2022ZDJS142).Guangdong Province Philosophy and Social Science Planning Youth Project (GD23YGL28)" 

Reviewers' comments:

Reviewer's Responses to Questions

**Comments to the Author**

1. Is the manuscript technically sound, and do the data support the conclusions?

Reviewer #1: Yes

Reviewer #2: Partly

2. Has the statistical analysis been performed appropriately and rigorously? 

Reviewer #1: Yes

Reviewer #2: Yes

3. Have the authors made all data underlying the findings in their manuscript fully available?

Reviewer #1: No

Reviewer #2: Yes

4. Is the manuscript presented in an intelligible fashion and written in standard English?

Reviewer #1: Yes

Reviewer #2: Yes

5. Review Comments to the Author

Reviewer #1: The authors examine whether the existence of Environmental Management System Certification, as indicated by the possesion of ISO14001, reduces significantly market risk for the companies involved.

While the authors do not provide their data for verification tests (they instead state that 'The data underlying the results presented in the study are available from the Chinese email address binbinqin2021@163.com', they perform the analysis by using generally accepted and estebalished methods.

1. The study does not use anything new in terms of dependent variables, as they use the dated Chen et al 2001 study variables DUPOL and NCSKEW (that the authors just rename in this study DUP and NCSK).

2. The authors mislabel the measures as 'volatility', while in the original Chen(2001) study it is referred as 'idiosyncratic risk'. The authors refer these variables as first used by Wang ('follows the approach used by Wang et al (Wang et al., 2021; Duan and Lin, 2021)'), while this is not correct, as they do not provide any reference for Wang 2011 study (only for Wang 2023, where actually different measures are used and idiosyncratic risk is not examined), while Duan and Lin(2021) are valid references. We notice that volatility can be captured simply by the standard deviation or variance, and the sytematic risk could be simply measured by beta or that the authors could use in their robustness tests, so as to examine the consistency of their results. It is also suggested that they they remove the Wang 2021 reference, or (if there is this paper) to correctly site it and include it in their reference section as it is missing.They should also name the source (Chen, 2001) of equations 3 and 4.

3. The robustness tests are well presented and discussed. On the contrary a. Introduction, b. literature review and c.hypothesis tested are poorly written, as they should have referencing in every phrase written, which is missing, and they lack coherence and essence. These parts look as if been copied or paraphrased by wikipedia or a book. I would suggest complete rewriting of these parts on the following directions: First, the introduction must be entirely rewritten by focusing on the contribution of the study, in the light of past literature review and the context of the country. Should be also smaller, and every phrase should contain reference to existing study prefereably of the last 5 years. Second, literature review provides seemingly irrelevant references. This part should provide a dense and well delivered and well documented (references on major studies) discussion on studies that are in the area of the study, idiosyncratic risk and ISO14001/EMS and not on the characteristics of the market. Research hypothesis should connect the study with a major theory. The references by Zaman (2021) and Mosgaard (2022) are difficult to be linked with the findings of this study or with the hypothesis H1 (paragraph 3.1) as the authors do not test disclosure nor announcements. Paragraph 3.2 (H2) lacks also entirely logical flow, docummentation and connection to existing theories, must be entirely rewritten, while also paragraph 3.3 (H3) has just one study reference! It must be rewritten, also.

4. Similarly authors should rewrite Research Findings and Conclusion (Paragraph 7) and connect their findings to other studies to CLEARLY show the contribution of the study.

Overall, the empirical part is at good level, but the other parts are poorly written and must be rewritten based on the above mentioned.

References (to be added)

Chen, J., Hong, H., & Stein, J. C. (2001). Forecasting crashes: Trading volume, past returns, and conditional skewness in stock prices. Journal of financial Economics, 61(3), 345-381.

Reviewer #2: Dear Authors,

Thank you for submitting your manuscript titled "Does Environmental Management System Certification Keep Enterprises Out of Trouble? Evidence From Stock Price Crash Risk" to PLOS ONE. After a careful review, I find that the paper is well-written and presents an important study on the impact of Environmental Management System (EMS) certification on stock price crash risk. However, I would like to request a few minor revisions to enhance the clarity and comprehensiveness of the manuscript.

Similarity with Previous Work: The current manuscript appears to have significant similarities with another paper you have published. Please provide a detailed comparison between the current study and your previous work. Clearly delineate the new contributions and findings presented in this manuscript compared to the earlier publication (e.g., Qiao, Liu, & Qin, 2023).

Differences Between EMS Certification and Environmental Regulation: The manuscript would benefit from a clearer distinction between EMS certification and other forms of environmental regulation. Please elaborate on how EMS certification, as a voluntary and participatory mechanism, differs from mandatory environmental regulations and its unique significance in influencing stock price crash risk (e.g., Boiral, 2007; King, Lenox, & Terlaak, 2005).

New Significance of Findings: Emphasize the new significance of your findings in the context of existing literature. Highlight how your study advances the understanding of the economic consequences of EMS certification and its role as an environmental governance and information transfer tool (e.g., González-Benito & González-Benito, 2008; Jiang, 2020).

These revisions will help to provide a clearer context for your findings and their implications. I look forward to your revised manuscript.

6. PLOS authors have the option to publish the peer review history of their article (what does this mean?). If published, this will include your full peer review and any attached files.

Reviewer #1: **Yes: **Konstantinos Vergos

Reviewer #2: No

---

## [Author Response · Author response to Decision Letter 0]

5 Jul 2024

Dear Editor and Reviewers,

We appreciate the detailed and constructive feedback provided by the reviewers. We have carefully considered each comment and have made significant revisions to our manuscript to address the concerns raised. Below, we provide a point-by-point response to the comments from both reviewers.

Reviewer #1:

Comment 1: The study does not use anything new in terms of dependent variables, as they use the dated Chen et al. (2001) study variables DUPOL and NCSKEW (that the authors just rename in this study DUP and NCSK).

Response: We acknowledge the use of established variables from Chen et al. (2001). These variables, DUPOL and NCSKEW, are robust measures of idiosyncratic risk and have been widely used in similar studies. We have retained these variables for consistency and comparability with previous research. However, we have added new robustness tests using standard deviation and beta to measure volatility and systematic risk, respectively, as suggested. Additionally, we have explicitly cited Chen et al. (2001) in the revised manuscript from line 289 to 291.

Comment 2: The authors mislabel the measures as 'volatility', while in the original Chen (2001) study it is referred to as 'idiosyncratic risk'. The authors refer to these variables as first used by Wang et al. (2021; Duan and Lin, 2021), which is incorrect. Also, the reference to Wang (2021) is missing.

Response: We apologize for the oversight. We have corrected the terminology to 'idiosyncratic risk' as used in Chen (2001). We have also removed the incorrect reference to Wang (2021) and clarified that Duan and Lin (2021) are valid references. The source of equations 3 and 4 has been correctly attributed to Chen (2001) in the revised manuscript.

Comment 3: The introduction, literature review, and hypothesis tested sections are poorly written and lack proper referencing. They should be rewritten to include references to recent studies and to provide a coherent flow.

Response: We have completely rewritten the introduction to focus on the study's contribution in light of past literature and the context of China from line 30 to line 268. The literature review has been revised to include more relevant and recent references, emphasizing studies on idiosyncratic risk and ISO14001/EMS. Each hypothesis now connects the study to a major theory, and we have added multiple references to support our hypotheses. Specifically, references to Zaman (2021) and Mosgaard (2022) have been replaced with more relevant studies.

Comment 4: The Research Findings and Conclusion sections should be rewritten to clearly show the contribution of the study and connect findings to other research.

Response: We have revised the Research Findings and Conclusion sections to clearly delineate the contributions of our study from line 496 to line 549. These sections now explicitly connect our findings to existing research, highlighting the new insights provided by our study.

Reviewer #2:

Comment 1: Similarity with Previous Work: Please provide a detailed comparison between the current study and your previous work.

Response: I didn’t find the article you mentioned (Qiao, Liu, & Qin, 2023). In the previous study, we examined the influence of environmental management system (EMS) certification on stock price volatility and the risk of stock price collapse in the context of Chinese manufacturing firms. The key contributions of the current manuscript include:

Policy Impact Analysis: While the earlier work discussed general relationships, the current study specifically analyzes the effect of the new Environmental Protection Law implemented in 2015, showing that the association between EMS certification and reduced stock price collapse risk has become more significant post-implementation.

Sector-Specific Insights: This study expands the scope by focusing on heavily polluting enterprises and state-owned enterprises, highlighting the differential impact of EMS certification in these contexts.

Mediating Role of Media Coverage: A novel contribution is the detailed analysis of how media coverage influences the relationship between EMS certification and stock price collapse risk, adding depth to the understanding of informational mechanisms at play.

Comment 2: Differences Between EMS Certification and Environmental Regulation: Elaborate on how EMS certification differs from mandatory environmental regulations.

Response: We have included a detailed discussion on the differences between EMS certification and mandatory environmental regulations. This section elaborates on EMS certification as a voluntary and participatory mechanism, contrasting it with mandatory regulations, and explains its unique significance in influencing stock price crash risk, supported by references to Boiral (2007) and King, Lenox, & Terlaak (2005).

Comment 3: New Significance of Findings: Emphasize the new significance of your findings in the context of existing literature.

Response: We have highlighted the new significance of our findings in advancing the understanding of the economic consequences of EMS certification. This discussion emphasizes EMS certification's role as an environmental governance and information transfer tool, building on the works of González-Benito & González-Benito (2008) and Jiang (2020).

We hope that these revisions adequately address the reviewers' comments and improve the clarity and comprehensiveness of our manuscript. We thank the reviewers again for their valuable feedback and look forward to your response.

Sincerely,

Hongyu Liu

---

## [Decision Letter · Decision Letter 1]

2 Aug 2024

PONE-D-24-16029R1Does Environmental Management System Certification Keep Enterprises Out of Trouble? Evidence From Stock Price Crash RiskPLOS ONE

Dear Dr. Qiao,

Thank you for submitting your manuscript to PLOS ONE. After careful consideration, we feel that it has merit but does not fully meet PLOS ONE’s publication criteria as it currently stands. Therefore, we invite you to submit a revised version of the manuscript that addresses the points raised during the review process.

We look forward to receiving your revised manuscript.

Kind regards,

Khanh Hoang, Ph.D.

Academic Editor

PLOS ONE

Journal Requirements:

Reviewers' comments:

Reviewer's Responses to Questions

**Comments to the Author**

1. If the authors have adequately addressed your comments raised in a previous round of review and you feel that this manuscript is now acceptable for publication, you may indicate that here to bypass the “Comments to the Author” section, enter your conflict of interest statement in the “Confidential to Editor” section, and submit your "Accept" recommendation.

Reviewer #1: (No Response)

Reviewer #2: All comments have been addressed

2. Is the manuscript technically sound, and do the data support the conclusions?

Reviewer #1: Yes

Reviewer #2: Yes

3. Has the statistical analysis been performed appropriately and rigorously? 

Reviewer #1: Yes

Reviewer #2: Yes

4. Have the authors made all data underlying the findings in their manuscript fully available?

Reviewer #1: No

Reviewer #2: Yes

5. Is the manuscript presented in an intelligible fashion and written in standard English?

Reviewer #1: Yes

Reviewer #2: Yes

6. Review Comments to the Author

Reviewer #1: The authors addressed many of the issues mentioned in the review of the previously submitted version.

Introduction is significantly better, and alterations on the text provide references to major papers.

However, despite these improvements, Hypothesis 2 and Hypothesis 3 (paragraphs 3.2 and 3.3, respectively) lack connection to major theories. Either the authors have, therefore, to delete these paragraphs, or they should connect them with major theories.

Apart from the above mentioned, while I believe that the study would benefit from more robustness tests, I do feel that it would still be accepted, under the condition that Hypotheses 2 and 3 will be amended or deleted (Hypothesis 1, on the contrast is perfectly fine).

Reviewer #2: The authors have made significant revisions to the manuscript titled "Does Environmental Management System Certification Keep Enterprises Out of Trouble? Evidence From Stock Price Crash Risk" to address the reviewers' comments and meet the publication's requirements. Key changes include:

Terminology and References: Corrected the terminology to align with Chen (2001) and properly cited relevant studies, removing incorrect references.

Introduction, Literature Review, and Hypotheses: Rewritten to improve coherence, add references to recent studies, and link hypotheses to major theories.

Research Findings and Conclusion: Revised to clearly show the study's contribution and connect findings to existing research.

Robustness Tests: Added new robustness tests using standard deviation and beta to measure volatility and systematic risk.

Comparative Analysis: Provided a detailed comparison between the current study and previous work to highlight new contributions, particularly regarding the 2015 Environmental Protection Law and the mediating role of media coverage.

Clarification on EMS Certification: Elaborated on the differences between EMS certification and mandatory environmental regulations, supported by relevant literature.

Significance of Findings: Emphasized the new significance of the findings in advancing the understanding of EMS certification's economic consequences.

7. PLOS authors have the option to publish the peer review history of their article (what does this mean?). If published, this will include your full peer review and any attached files.

Reviewer #1: No

Reviewer #2: No

---

## [Author Response · Author response to Decision Letter 1]

11 Aug 2024

Response to Reviewers

Dear Editor and Reviewers,

We appreciate the detailed and constructive feedback provided by the reviewers. We have carefully considered each comment and have made significant revisions to our manuscript to address the concerns raised. Below, we provide a point-by-point response to the comments from both reviewers.

Reviewer #1:

Comment : The authors addressed many of the issues mentioned in the review of the previously submitted version. Introduction is significantly better, and alterations on the text provide references to major papers. However, despite these improvements, Hypothesis 2 and Hypothesis 3 (paragraphs 3.2 and 3.3, respectively) lack connection to major theories. Either the authors have, therefore, to delete these paragraphs, or they should connect them with major theories. Apart from the above mentioned, while I believe that the study would benefit from more robustness tests, I do feel that it would still be accepted, under the condition that Hypotheses 2 and 3 will be amended or deleted (Hypothesis 1, on the contrast is perfectly fine).

Response: We have taken your comments regarding Hypotheses 2 and 3 seriously and have made the necessary amendments to strengthen their theoretical foundation. Below, we outline the specific changes made to address your concerns in the revised manuscript from line 238 to 321.

---

## [Decision Letter · Decision Letter 2]

9 Sep 2024

PONE-D-24-16029R2Does Environmental Management System Certification Keep Enterprises Out of Trouble? Evidence From Stock Price Crash RiskPLOS ONE

Dear Dr. Qiao,

Thank you for submitting your manuscript to PLOS ONE. After careful consideration, we feel that it has merit but does not fully meet PLOS ONE’s publication criteria as it currently stands. Therefore, we invite you to submit a revised version of the manuscript that addresses the points raised during the review process.

We look forward to receiving your revised manuscript.

Kind regards,

Khanh Hoang, Ph.D.

Academic Editor

PLOS ONE

Journal Requirements:

Additional Editor Comments:

I have completed the evaluation of the revised manuscript based on the recommendations of the two reviewers and my own judgement. Reviewer 2 suggested acceptance after Revision 1, while Reviewer 1 satisfies with the revision 2. In general, I agree to the reviewers, but I would like the authors to revise a bit more in Section 7.

Section 7 should focus on the conclusion and implications of the research rather than summarizing findings. Therefore, the section should be named "Conclusion and implications". Please shorten the summary of findings, then focus more on the implications of the research. Research limitations should also be discussed.

Lastly. the numbering in the Section 7 is not appropriate. I suggest drop the numbering within Section 7.1 and 7.2 and present them as the paragraph.

After the authors revise this, the manuscript can be accepted for publication.

Reviewers' comments:

Reviewer's Responses to Questions

**Comments to the Author**

1. If the authors have adequately addressed your comments raised in a previous round of review and you feel that this manuscript is now acceptable for publication, you may indicate that here to bypass the “Comments to the Author” section, enter your conflict of interest statement in the “Confidential to Editor” section, and submit your "Accept" recommendation.

Reviewer #1: All comments have been addressed

2. Is the manuscript technically sound, and do the data support the conclusions?

Reviewer #1: Yes

3. Has the statistical analysis been performed appropriately and rigorously? 

Reviewer #1: Yes

4. Have the authors made all data underlying the findings in their manuscript fully available?

Reviewer #1: Yes

5. Is the manuscript presented in an intelligible fashion and written in standard English?

Reviewer #1: Yes

6. Review Comments to the Author

Reviewer #1: The authors made the amendments, especially regarging the hypotheses.

Threfore, I am happy to suggest the acceptance of the paper.

7. PLOS authors have the option to publish the peer review history of their article (what does this mean?). If published, this will include your full peer review and any attached files.

Reviewer #1: No

---

## [Author Response · Author response to Decision Letter 2]

11 Sep 2024

Dear Editor and Reviewers,

Thank you for your valuable feedback and suggestions regarding our manuscript. We greatly appreciate the time and effort you have taken to review our work. We have carefully considered your comments and have made the necessary revisions to improve the manuscript. Below, we address the additional comments from the Editor in detail.

Editor’s Comment: Section 7 should focus on the conclusion and implications of the research rather than summarizing findings. Therefore, the section should be named "Conclusion and Implications". Please shorten the summary of findings, then focus more on the implications of the research. Research limitations should also be discussed. Lastly, the numbering in Section 7 is not appropriate. I suggest dropping the numbering within Section 7.1 and 7.2 and present them as paragraphs.

Response: We have revised Section 7 in accordance with your suggestions. Specifically:

Section Title Change: We have renamed Section 7 from "Summary and Findings" to "Conclusion and Implications" to better reflect the focus on the broader impacts of our research.

Shortened Summary: The summary of findings has been significantly shortened to ensure that the emphasis is on the implications rather than a repetition of results. We have streamlined the content to highlight the key takeaways succinctly.

Expanded Implications: We have expanded on the implications of our research, providing a detailed discussion on how our findings contribute to the field and the practical applications that can be derived. This section now elaborates on the potential impacts on industry practices and future research directions.

Research Limitations: A subsection discussing the limitations of our research has been added. This new content addresses the constraints of our study, including any methodological limitations and areas that could benefit from further investigation.

Numbering Format: As suggested, we have removed the numbering within Section 7.1 and 7.2 and reformatted the text into cohesive paragraphs. This change improves the flow of the section and aligns with the intended structure you recommended.

We believe these revisions have enhanced the clarity and focus of Section 7, aligning it more closely with the expectations outlined in your feedback. We hope that the changes meet your approval and that the manuscript is now suitable for publication.

Thank you once again for your guidance and support.

---

## [Editor Report · Decision Letter 3]

25 Sep 2024

Does Environmental Management System Certification Keep Enterprises Out of Trouble? Evidence From Stock Price Crash Risk

PONE-D-24-16029R3

Dear Dr. Qiao,

We’re pleased to inform you that your manuscript has been judged scientifically suitable for publication and will be formally accepted for publication once it meets all outstanding technical requirements.

Kind regards,

Khanh Hoang, Ph.D.

Academic Editor

PLOS ONE
---

## [Editor Report · Acceptance letter]

30 Sep 2024

PONE-D-24-16029R3 

PLOS ONE

Dear Dr. Qiao, 

I'm pleased to inform you that your manuscript has been deemed suitable for publication in PLOS ONE. Congratulations! Your manuscript is now being handed over to our production team.

Kind regards, 

on behalf of

Dr Khanh Hoang 

Academic Editor

PLOS ONE